# Can Artificial Intelligence Improve the Management of Pneumonia

**DOI:** 10.3390/jcm9010248

**Published:** 2020-01-17

**Authors:** Mariana Chumbita, Catia Cillóniz, Pedro Puerta-Alcalde, Estela Moreno-García, Gemma Sanjuan, Nicole Garcia-Pouton, Alex Soriano, Antoni Torres, Carolina Garcia-Vidal

**Affiliations:** 1Infectious Diseases Department, Hospital Clínic of Barcelona, 08036 Barcelona, Spain; marianachumbita0504@gmail.com (M.C.); pedro.puerta84@gmail.com (P.P.-A.); emorenog@clinic.cat (E.M.-G.); gsg7546@gmail.com (G.S.); nicole.garciapouton@gmail.com (N.G.-P.); asoriano@clinic.cat (A.S.); 2Institut d’Investigacions Biomèdiques August Pi i Sunyer (IDIBAPS), 08036 Barcelona, Spain; catiacilloniz@yahoo.com (C.C.); atorres@clinic.cat (A.T.); 3Department of Pneumology, Institut Clinic del Tórax, Hospital Clinic of Barcelona, SGR 911-Ciber de Enfermedades Respiratorias (Ciberes), 08036 Barcelona, Spain; 4School of Medicine, University of Barcelona, 08036 Barcelona, Spain

**Keywords:** artificial intelligence, pneumonia

## Abstract

The use of artificial intelligence (AI) to support clinical medical decisions is a rather promising concept. There are two important factors that have driven these advances: the availability of data from electronic health records (EHR) and progress made in computational performance. These two concepts are interrelated with respect to complex mathematical functions such as machine learning (ML) or neural networks (NN). Indeed, some published articles have already demonstrated the potential of these approaches in medicine. When considering the diagnosis and management of pneumonia, the use of AI and chest X-ray (CXR) images primarily have been indicative of early diagnosis, prompt antimicrobial therapy, and ultimately, better prognosis. Coupled with this is the growing research involving empirical therapy and mortality prediction, too. Maximizing the power of NN, the majority of studies have reported high accuracy rates in their predictions. As AI can handle large amounts of data and execute mathematical functions such as machine learning and neural networks, AI can be revolutionary in supporting the clinical decision-making processes. In this review, we describe and discuss the most relevant studies of AI in pneumonia.

## 1. Introduction

According to the Global Burden of Diseases, Injuries, and Risk Factors Study (GBD) 2017, pneumonia remains the leading cause of death, especially among children younger than five years of age and adults older than 70 years of age. Despite the fact that there was a reduction in deaths from pneumonia of 36% between 2007 and 2017 in children under the age of five, the rate has risen by 34% among older adults (≥70 years) [1]. During the last few decades, the amount of pneumonia patients requiring intensive care management has increased overall, with the emergence of multidrug- resistant (MDR) pathogens making pneumonia management increasingly difficult for physicians [2,3,4].

The use of artificial intelligence (AI) to support clinical decision-making processes is an appealing concept. However, most clinical physicians do not understand the degree of practicality to which AI can serve in current medical practice. Two factors highlight the connection between AI and routine medical practice: the availability of data from electronic health records (EHR) and advances made in computational performance. The latter allows for both extensive mathematical operations and the training of a high quantity of data through different mathematical approaches such as machine learning (ML) or neural network (NN) models in rapid time (Figure 1).

Nowadays, there exists a high number of variables related to patient care and medical history, rendering the understanding of what is occurring to patients to be complex. As medical research continues to expand rapidly, it is extremely difficult to be up-to-date with the most recent advances made in medicine. A recent publication estimated that by 2020, there will be 200 times more medical information than what a single individual would be able to read in his/her entire life [5]. This accumulation of large amounts of data will occur naturally alongside challenges in medicine that include a growing population lifespan and an increase in complex medical history. However, current clinical practices observe physicians making decisions based on personal and medical team experience, as well as study analyses conducted with minimal data or treatment guidelines. Should physicians desire to address the advent of difficulties more effectively and efficiently, then it may be worth considering the inclusion of AI in medical practice.

In conjunction with medical literature, the introduction of mathematical models and NN could allow for the ordering and verification of all data retrieved from electronic health records (EHRs) and facilitate mass improvements in patient care. Treatment personalization strategies or site of care could be optimized as a result of predicting clinical situations that could deteriorate patients’ health, i.e., risks of multidrug-resistant infections, dangers of potential treatments, risks of readmissions, etc.

## 2. Aims and Limitations

The focus of this review is to explain the potential of AI in improving the clinical decision-making processes, making note of our current experience in the area and framing the discussion around patients with pneumonia. We summarized the most relevant evidence of AI for the management of pneumonia.

## 3. Some Relevant Data about Revolution AI in Medicine

As previously stated, two important revolutions will transform the landscape of our current research and clinical approaches. The first revolution relies on the massive availability of data; that is, a high amount of data can be retrieved from EHRs in real time. This aspect contrasts with former ways of collecting data, which required traditional clinical recorders [6,7].

Currently, most hospitals have EHRs; indeed, more than three quarters of hospitals in US had adopted at least a basic EHRs [8]. However, the concern held worldwide is that of their complex storage structure. EHRs come from different sources. Accessibility to those records is limited, making large datasets difficult to construct. Yet, this challenge can be overcome with the proper tool, provided that it allows for the storage and analysis of large and complex datasets through “4 v”: volume, variety, velocity, and value [9]. Our team has observed that the development of such an intelligent system to read and retrieve data from EHRs has led to the retrieval of more than 850 million pieces of data from our EHRs for current potential analyses. This breakthrough is remarkable, as it has shaped how our team performs epidemiological and comparative studies.

The second revolution is the improved performance of computers. Currently, computer systems operate at high speeds and possess a great capacity for handling information in short time. Through ML and NN, these computer-driven mathematical approaches can use the massive clinical information we store and predict different clinical situations.

In our experience, NN has proven to be significantly more accurate when training a model to predict several clinical scenarios.

To understand the efficacy of NN, it is important to see what it comprises. NN is a technique based on layers of artificial neurons. Each node (neuron) receives an input from an external source. Each one then has a value or weight assigned and could later undergo modifications in the training phase of the model. This node is later processed through a network of connected, hidden nodes—each one possessing a specific function to eventually produce output results or in this particular context, predictions. In other words, the NN learns a model of data (training phase) and then given what it has learned, a data model can be used to classify new data (testing phase) [9]. The most attractive feature is its capacity for nonlinear analysis, provided that the linear method has been used traditionally; nevertheless, while appealing, it is not the best as a result of the many factors involved in patient care [10].

## 4. Our Experience in Artificial Intelligence Supporting Clinical Decision-Making Processes

With the 850 million pieces of data retrieved from our EHRs, we are able to identify a clinical problem through computer systems, collect data from EHRs and algorithm evaluations in optimal time, perform real-time predictions, and link such predictions to in-hospital clinical recommendations for improved clinical decision-making processes. Our innovative program has primarily focused on predicting the presence or absence of multidrug-resistant pathogens causing infections [11]. Such an approach to pneumonia management is currently being explored.

## 5. Current Research on Artificial Intelligence in Pneumonia

There is little information regarding the use of AI in improving the management of pneumonia (Table 1). Our search terms included AI and pneumonia. From a total of 103 we chose those in which AI proved helpful in diagnosing and managing pneumonia, and those whose provide data about AI method used. Until now, most AI was focused on pneumonia diagnosis via the study of chest X-rays (CXR) patterns. This fact is easy to understand, given that AI has great application potential in medical imaging where data is easier to obtain. Promising results in several areas of medical image analyses have been recently published [12], highlighting the detections of diabetic retinopathy in fundus photographs and metastasis in pathological images, as well as the classification of skin cancer from skin photographs. In contrast to unitary images examined by clinicians, image processing performed by AI tools allows for an image’s architecture to be broken down into millions of pixels for more precise discrimination.

## 6. AI for Pneumonia Diagnosis

Pneumonia symptoms and signs may be subjective and non-specific; therefore, a CXR is mandatory for diagnosis. Optimal interpretation of CXR is key for pneumonia diagnosis and might also help differentiate between different etiologies of pneumonia. In general, bacterial pneumonia typically exhibits a focal lobar consolidation, whereas viral pneumonia manifests as a more diffuse ‘‘interstitial’’ pattern in both lungs. However, CXR interpretation varies among physicians, and oftentimes there are other associated respiratory pathologies that complicate and further delay an appropriate diagnosis. In both elderly patients and patients with multiple comorbidities (cardiac and pulmonary), there exists the possibility of a misdiagnosis of pneumonia because of the difficulty posed in interpreting CXR.

In 2018, Kermany et al. [13] developed a NN algorithm to predict pneumonia diagnoses in a pediatric population. Viral versus bacterial pneumonia was distinguished as a secondary outcome to make an appropriate and rapid intervention. Images performed as part of clinical routine care were selected from a retrospective cohort of patients from two centers in Guangzhou, China. A total of 5232 chest images were used (3883, CXR showed pneumonia of which 2538 were bacterial pneumonia and 1345 viral pneumonia; the rest (1349) were normal CXR). Their AI algorithm achieved an accuracy rate for pneumonia detection of 92.8%, with a sensitivity of 93.2% and a specificity of 90.1%. The area under the receiver operating characteristic curve, (ROC) for pneumonia diagnosis was 96.8%. The test accuracy rate in differentiating bacterial and viral pneumonia was 90.7%, with a sensitivity of 88.6% and a specificity of 90.9%. The achieved area under the ROC curve was 94%. No comparison of the model with medical physicians was reported.

More recently, Stephen et al. [14] used NN to predict pneumonia from data provided by a CXR image dataset, which contained a total of 5856 CXR images of pediatric patients aged one to five years, with and without pneumonia (CXR imaging was performed as part of patients’ routine medical care). The authors employed data augmentation methods to artificially increase the size and quality of the dataset. Augmentation methods were created by restoring different characteristics, including a 1/255 scale and implementing changes, such as rotation range, width, height, cut, and zoom range. For example, rotation range adjustments were carried out by randomly rotating images by 40 degrees. The other aspects, that is, width, height, etc. underwent a variation of 0.2% each. The results were as followed: a training accuracy of 0.9531 and a validation accuracy of 0.9373. The study was not validated in humans.

Using a different approach, Heckerling et al. [15] tried to predict the presence or absence of pneumonia with a NN, using demographic characteristics, symptoms, signs, and comorbidity data obtained in two US centers. The specific data used were not reported. Patients with missing data were excluded, but the number and characteristics of patients excluded for this reason are not reported. One thousand and twenty-three patients were included, with all of them coming from one of the hospitals in the training cohort. The NN algorithms achieved a sensitivity of 0.466 and a specificity of 0.972 for pneumonia diagnosis, with an area under the ROC curve of 0.874. When applied to the testing cohort, that is, the 116 patients from the second center, a sensitivity of 0.317 and a specificity of 0.973 for pneumonia diagnosis were reported. There was no difference in ROC area between training and testing cohorts. The authors compared NN prediction with a pneumonia prediction obtained by logistic regression (ROC area of 0.87) and demonstrated a greater discriminatory accuracy for NN.

The most important study documenting an automated system classifying CXR was published recently by Hwang et al. [16]. The authors developed a deep-learning based NN algorithm that classified results from chest radiographs into major thoracic diseases: pulmonary malignant neoplasm, active tuberculosis, pneumonia, and pneumothorax. These authors used 54,221 normal CXR and 35,613 CXR with abnormal results from a single institution to train a deep-learning network. CXR were reviewed and categorized into different diseases by at least one of 15 board-certified radiologists with more than seven years of experience. CXR were randomly assigned into one of the three following datasets: (1) training data, (2) tuning data, and (3) in-house validation data to predict the different thoracic diseases. They also used four external hospitals as a validation cohort. The main result for in-house validation was an area under the ROC of 0.965 (95% CI, 0.955–0.975) for image-wise classification. Operating thresholds were defined as probabilities of 0.16 (high sensitivity threshold; sensitivity, 0.951; specificity, 0.750) and 0.46 (high specificity threshold; sensitivity, 0.920; specificity, 0.950). In the external validation analyses, the area under the ROC curve was 0.979 (0.973–1.000), even better than the in-house results. In this study, a comparison was also performed between computer algorithm results and physicians. For this study, an observer panel of 15 physicians with varying experience (thoracic board-certified radiologists; and five non-radiology physicians) was compiled. These physicians evaluated data from the external validation data set. Two sessions were held: in the first, physicians examined every CXR without any assistance; in the second, physicians were able to use the AI algorithm to support decision-making processes and modify their original decision if deemed necessary. In session 1, pooled area under the ROC curve was 0.814, 0.896, and 0.932 for non-radiology physicians, board certified radiologists, and thoracic radiologists, respectively. The deep-learning NN algorithm was significantly better than all three observer groups (all *p* < 0.005). In session 2 of the observer performance test, with the assistance of the AI algorithm, area under the ROC curve of non-radiology physicians, board-certified radiologists, and thoracic radiologists was 0.904, 0.939, and 0.958, respectively. Increments of area under ROC were 0.090, 0.043, and 0.026, respectively, and all were statistically significant (all *p* < 0.005). The authors concluded that the algorithm improved the quality and efficiency of thoracic disease recognition.

All these studies demonstrate the importance of artificial intelligence in the diagnosis of pneumonia using radiographic patterns. However, it is still necessary to perform validations of these studies to be able to use them as part of routine clinical practice.

## 7. Empirical Therapy

The decision-making process for empirical antibiotics in lower respiratory tract infections is a challenge, especially in an elderly population who more commonly present with atypical symptoms, higher comorbidities, a lower likelihood in identifying the causing infection agent, and higher mortality.

Gueli et al. [17] developed a NN model to help decide the most effective empirical antibiotic in a population with acute symptoms and without microbiological samples. They included 117 patients aged between 55 and 97, who were hospitalized in the geriatric ward of an Italian hospital with a diagnosis of either pneumonia, exacerbated chronic obstructive pulmonary disease (COPD), or bronchopneumonia with respiratory failure. The authors evaluated 11 clinical variables (age, heart problems, smoking habit, antibiotics, etc.) reported in the clinical history to predict antibiotic treatments with optimal efficacy. The model consisted of a 105-episode training dataset and a 23-episode validation dataset. Authors applied an NN model that chose antibiotics in 20 patients and compared such group to another 20 subjects whose therapy was determined by physicians. The groups were compared per therapy efficacy, mean duration of therapy, and hospitalization. Authors found that the NN model could provide positive contributions when choosing empirical antibiotics for acute pulmonary infections.

Another aspect that has been studied is the use of AI for mortality prediction in community- acquired pneumonia (CAP) patients. Ward et al. [18] retrospectively assessed 4531 patients from data collected prospectively in Hospital Clinic of Barcelona. They evaluated a casual probabilistic network (CPN) previously used for the purpose of predicting mortality in patients with bacteremia and extended it to predict the probability of death within 30 days in patients with pneumonia. They used Sepsis Finder CPN model and compared with traditional scores (Pneumonia Severity Index –PSI-, CURB 65-score, Sequential Organ Failure Assessment –SOFA-, quick SOFA -q-SOFA-). The area under curve (AUC) in the ROC curve was 0.88 for Sepsis Finder, better than CURB-65 (0.759, *p* < 0.001), SOFA (0.670, *p* < 0.001), and q-SOFA (0.642, *p* < 0.001). 

Recently, this model was validated in data gathered from 1034 CAP patients from the Hospital La Fe in Valencia, Spain. The AUC for Sepsis Finder CPN model was 0.803, which was similar to that of the previous study (0.811). The AUC for the validation data was not significantly different to that for PSI (AUC 0.830, *p =* 0.42) and CURB-65 (AUC 0.763, *p =* 0.20). Future studies are needed in order to generalize this model [19].

## 8. Conclusions

AI could facilitate screening programs and create more efficient referral systems in all of medicine. The impact of such advances could lead to breakthroughs in clinical and public health. As it currently stands, there exist both a large amount of EHR data and different mathematical approaches, such as ML or NN, to support the use of AI in aiding clinical decision-making processes.

In pneumonia care, many AI programs have focused on performing early radiological diagnoses. Most studies have employed NN and observed high accuracy rates in their predictions. According to the only study comparing computers with physicians in diagnosing pneumonia, AI algorithms improved diagnosis overall.

There has yet to be applications of AI algorithms in areas of pneumonia management, such as site of treatment, empirical antibiotic decisions, prediction of mechanical ventilation need, or outcome stratification. The potential usefulness of such inclusion will be explored most likely within the near future.

## Figures and Tables

**Figure 1 jcm-09-00248-f001:**
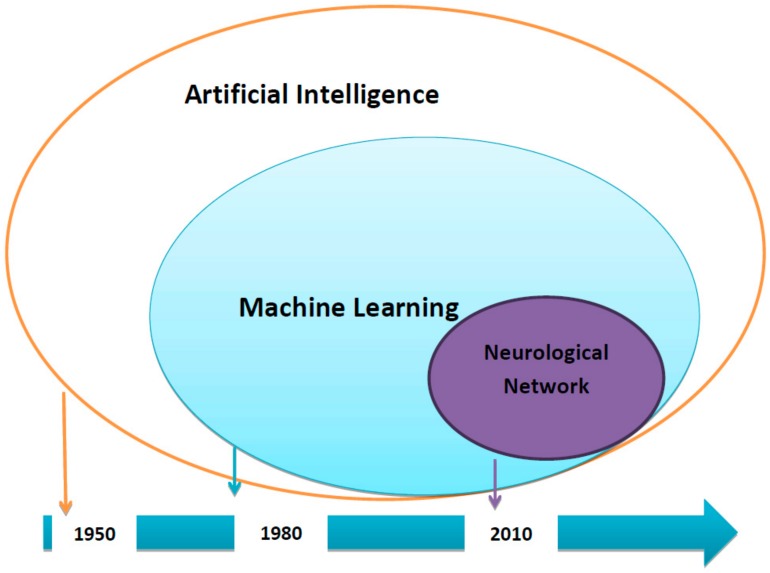
Time-Line of Artificial Intelligence.

**Table 1 jcm-09-00248-t001:** Recently published AI approaches being undertaken to support clinical decision-making processes in pneumonia.

Reference	Kermany et al. [13] *Cell,* 2018	Stephen et al. [14] *Journal of Healthcare Engineering*, 2019	Heckerling et al. [15] *Clinical Applications,* 2003	Hwang et al. [16] *JAMA Network Open,* 2019
Main Goal	Detect pneumonia and distinguish viral and bacterial etiology	To handle pneumonia classification	Predict the presence of pneumonia among patients with acute respiratory complaints	Make a deep learning–based algorithm for major thoracic diseases; Comparison with physicians and external validation
Applied Method	Neural network	Neural network and augmentation methods to artificially increase the size and quality of the dataset	Neural networks	Deep learning—neural networks
N°	5232 chest X-ray for training phase and 624 images for test phase	5856 X-ray images—3722 training set and 2134 to the validation set---	1023 patients–training cohort of 907 and a testing cohort of 116	54,221 X-ray with normal finding—41140 with abnormal findings
Results	Detect pneumonia = accuracy of 92.8% Distinguish viral vs bacterial = accuracy of 90.7%	Training accuracy = 0.9531 validation accuracy of 0.9373	Training cohort = sensitivity of 0.842 specificity of 0.593 testing cohort = sensitivity of 0.829 specificity of 0.547	Image-wise classification: in-house = AUROC of 0.965 and external validation = AUROC of 0.979 Lesion-wise localization: in-house = AUAFROC of 0.916 and external validation = AUAFROC of 0.972 -Comparison with physician: DLAD = AUROC 0.983 was higher versus 3 observer groups (*p* < 0.005)

Abbreviations: AUROC: area under the receiver operating characteristic curve; AUAFROC: area under the alternative free-response receiver operating characteristic curve; DLAD: Deep learning–based automatic detection algorithms.

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
