# Peer review of "Can Artificial Intelligence Improve the Management of Pneumonia"

_jcm, 2020, doi:10.3390/jcm9010248_

Round 1

Reviewer 1 Report

Authors addressed all my comments sufficiently. I accept the manuscript for publication 

Reviewer 2 Report

*

Reviewer 3 Report

Dr. Chumbita M, et al. wrote the review article about the AI probability for pneumonia medicine. This field of medicine is the very interesting and hot area. This paper will draw attention of many readers. The manuscript is well described and understandable. I believe this review will make impact on physicians.

This manuscript is a resubmission of an earlier submission. The following is a list of the peer review reports and author responses from that submission.

Round 1

Reviewer 1 Report

The aim of the current study is well intended and an interesting one but the clarity is masked by 1) poor english phrasing and 2) the statement that the authors' profession as clinicians would somehow be a valid excuse for much too simplified review. This is also stated on line 196. I would encourage the authors to engage in crossprofessional collaboration in order to improve the review to help deepen the understanding of the reader.

The reviewed articles are pleasantly described in the table the text but would benefit from a more stringent description. How did the authors find the articles to include in the review? Which search terms were used? No numbers of the amount of articles found is included.

Do the authors have any ambitions of their own to contribute to the field with an original study? If so it could be described in greater detail.

The reader would benefit from a description of the augmentation methods described in line 142.

Minor details

Line 6 There is a space after the last author, creating an uncertainty of whether a name is missing.

Line 34: the sentence describing a reduction in deaths due to pneumonia is unclear- what population, for what reason?

Line 37: the font appears to be larger

Line 64-67: this is an awkward section. If it is to remain in the manuscript I would urge the authors to include a heading such as "Aims and Limitations"

Line 84: consider something less literary than "at lightning speed". Also Line 108.."the following lines"

Line 97: "it is not the best" is an awkward phrasing- it should be described in a different more scientific manner

Line 104: predicting WHAT in multidrug-resistant infections?

Line 144: humans

Line 226: the first page indicates the 2 first authors as equally contributing, but here it says that ALL authors contributed equally.

Reviewer 2 Report

Thank you for the possibility for the review of this interesting paper. Nowadays, techniques involving AI are under intensive assessment and the results are promising and encouraging. 
The manuscript is a curious and well organized summary of the role of AI in diagnosis of pneumonia. The didactic value of the work is also high and presents contemporary level of knowledge, based on well chosen references. 
The only one issue which I want to raise to discuss is Author Contributions section - “All the authors contributed equally to this article”, while in the title page only two of authors were marked as equally contributing. Besides this, equal contribution with such number of authors and character of the study is almost impossible - please, correct this statement.

English grammar and style would be also improved in some areas of the text.

After addressing above mentioned issues, I recommend the manuscript to be published in the journal.